# At the Crossroads between Eating Disorders and Body Dysmorphic Disorders—The Case of Bigorexia Nervosa

**DOI:** 10.3390/brainsci13091234

**Published:** 2023-08-24

**Authors:** Octavian Vasiliu

**Affiliations:** Department of Psychiatry, Dr. Carol Davila University Emergency Central Military Hospital, 010816 Bucharest, Romania; octavvasiliu@yahoo.com

**Keywords:** eating disorders, muscle dysmorphia, bigorexia nervosa, body dysmorphic disorder, anabolic steroids abuse, physical exercise addiction, schizophrenia spectrum disorders

## Abstract

Bigorexia nervosa (BN) is a controversial nosological entity, considered either a feeding/eating disorder (FED) or a subtype of body dysmorphic disorder (BDD). This rapid review aims to explore the characteristic features of BN and identify evidence-based therapeutic interventions for this condition. Three electronic databases (PubMed, Cochrane, and Google Scholar) were searched for relevant information about BN, and 26 reports were reviewed in detail. The results showed that bodybuilders, weightlifters, and other populations involved in athletic activities are the most vulnerable to the onset of this disorder. Patients with BN should also be screened for physical and psychiatric comorbidities and complications, such as anabolic steroid use disorder, physical exercise addiction, and depressive or anxiety disorders. The main differential diagnoses for BN are schizophrenia spectrum disorders, depressive disorders, anxiety disorders, bodily distress disorder, and obsessive–compulsive disorders. Using validated screening instruments is considered very important from a clinical perspective, with the aim of providing early identification of this disorder. Therapeutic interventions for patients with BN are still in the early phases of development, and no specific pharmacological treatment has yet been identified. Since it is similar to the obsessive–compulsive spectrum, cognitive behavioral therapy has been suggested as a useful intervention; however, it has not yet been validated in large-scale clinical trials. In conclusion, based on the reviewed data, clarifying the concept of BN is of practical importance for constructing adequate prevention strategies and validating proper therapeutic interventions.

## 1. Introduction

Feeding and eating disorders (FEDs) represent a major cause of disability in adolescent girls and young adult women, whereas, in men, they are less frequently reported [1,2]. The pathogenesis of FEDs is still poorly understood, but both genetic (e.g., ≥50% of the variance in vulnerability to FEDs and disordered eating behaviors can be explained by additive genetic effects) and environmental factors (e.g., Western cultural ideals of appearance, perfectionistic attitude toward the body, and dysfunctional patterns of familial interaction) have been explored [1,2,3,4,5].

Although the vast majority of data on FEDs are derived from studies enrolling female patients, evidence exists that males are affected by this type of disorder with equally severe clinical symptomatology [6,7,8]. The possibility that epidemiological data may be biased due to the under-reporting of data from male patients or different forms of presentations (i.e., muscularity-focused body image concerns, depression and shame, comorbid substance use disorders, and sexual orientation) cannot be overlooked [9,10]. For example, in a sample of 135 males presenting with FEDs, more than 40% presented with bulimia were identified as homosexual or bisexual, whereas 58% with anorexia were asexual [10]. Also, it was suggested that men with “undifferentiated” or “feminine” gender roles were more vulnerable to the onset of disordered eating behaviors than men with “masculine” or “androgynous” roles [9,11]. In addition, media pressure for an ideal muscular body in men has been invoked as a factor that increases body dissatisfaction, excessive use of steroids, and strenuous physical exercise, and is a potential mediator for FED onset [9,12]. 

Bigorexia nervosa, or muscle dysmorphia, was initially reported in male bodybuilders, who continuously ruminated about their body mass and being larger or more muscular [13,14]. If obsessions are clearly related to inadequate body shape, compulsions refer to overexercising in the gymnasium, overbuying sports supplements, dysfunctional eating behavior, or substance use disorders (SUDs) [13]. Muscle dysmorphia is synonymous with “reverse anorexia”, first described in a study in the 1980s [14]. Also, the term “Adonis complex”, based on the name of the Greek god who represented a standard of masculinity, has been vehiculated as referring to an excessive focus on men’s body image [15]. 

These individuals considered themselves small and weak, although the reality was completely different, and they tended to avoid social gatherings because of fear they would be seen as too small [14]. They wore heavy clothes in the summer owing to their self-image distortions. This pathology was associated with significant functional consequences and up to three times higher suicide rates than other types of body dysmorphic disorders (BDDs) [16]. Poor quality of life and a high frequency of SUDs and anabolic steroid use were more frequently reported in patients with muscle dysmorphia than in the control group (N_1_ = 14 men with bigorexia vs. N_2_ = 49 with BDDs) [16].

Patients with bigorexia nervosa share clinical features with individuals diagnosed with anorexia nervosa, such as concerns about body weight and shape, appearance intolerance, and functional impairment [17,18], and disordered eating has been associated with a higher risk of bigorexia [19,20]. More than 20% of these patients had a history of anorexia nervosa, and almost 30% had a past diagnosis of any FED [19,20]. However, current nosological classifications support the inclusion of bigorexia nervosa in the category of “BDDs” [21,22].

As a consequence of this contradiction in the placement of bigorexia nervosa in the category of BDDs by two of the most influential classifications of mental disorders and the clinical and epidemiological data supporting the similarity between bigorexia nervosa and FEDs, a rapid review of available reports on this specific pathology was considered opportune. Also, based on the preliminary sources cited, other controversial aspects regarding bigorexia nervosa have been detected, such as the characterization of “reverse anorexia” as an exclusively male disorder (because of the “Adonis complex” concept), which requires an investigation of the available epidemiological data regarding this entity’s gender distribution; the observation of SUDs and bigorexia nervosa comorbidity, which raises the problem of causality and/or common risk factors; the influence of socio-cultural factors (i.e., through the concept of an ideal body shape) vs. vulnerability factors on the pathogenesis of bigorexia nervosa; and, last but not least, the characterization of the most vulnerable populations to the onset of bigorexia nervosa, which may lead to the creation of policies focused on prophylaxis and early intervention.

## 2. Objective and Methods

The main objective of this rapid review was to identify relevant data regarding the risk factors, diagnostic criteria, epidemiology, pathophysiology, structured evaluation, and treatment of bigorexia nervosa. The secondary objective was to determine whether clinical recommendations could be made, starting with the data retrieved from the literature.

Three major electronic databases (PubMed, https://pubmed.ncbi.nlm.nih.gov/ (accessed on 2 April 2023), Cochrane, https://www.cochrane.org/ (accessed on 4 April 2023), and Google Scholar, https://scholar.google.com/ (accessed on 5 April 2023) were searched using the paradigm “bigorexia nervosa” OR “bigorexia” OR “muscle dysmorphia” OR “reverse anorexia” AND “prevalence” OR “incidence” OR “risk factors” OR “diagnosis” OR “pathophysiology” OR “evaluation” OR “treatment”. The lists of references for each retrieved article were manually searched if considered relevant. The detailed inclusion/exclusion criteria are specified in Table 1. No language restrictions were implemented, and all types of available sources (primary and secondary reports) were reviewed if they contained relevant data for the previously mentioned objectives. The upper limit of the search interval was March 2023, with no inferior time limit.

## 3. Results

After applying the paradigm search, a total number of 4317 papers surfaced, but only 2858 remained after de-duplication. Succeding the application of the inclusion/exclusion criteria, out of the screened papers, only 23 reached the final phase of the review. Another 92 references were explored after the lists of references were consulted, but only 3 were considered for review after operational criteria were applied (Figure 1). 

Thus, a number of 26 **in extenso** papers were reviewed in detail, representing six reports on risk factors, ten on positive and differential diagnosis, five on epidemiological data, three on structured evaluation, and five on treatment (with a degree of overlap between sources depending on the pre-determined outcomes) (Table 2).

### 3.1. Risk Factors, Pathophysiology, and Comorbid Conditions in Bigorexia Nervosa

Although the discussion on the predisposing, precipitating, or perpetuating roles of anabolic androgenic steroid (AAS) administration in muscle dysmorphia is far from conclusive, the association between AAS use and bigorexia nervosa is clearly supported by evidence [23]. The use of AAS was significantly correlated with the presence of bigorexia nervosa at a statistical level, and patients stated that they decided to consume these substances because of their reverse anorexic symptoms [14]. Individuals who abused AAS presented with more severe muscle dysmorphia and assumed more rigid conceptions of conventional male roles [24]. Additionally, steroid-using bodybuilders were inclined to attain the ideal image of an exaggerated mesomorphic phenotype by any means [25].

A systematic review of this subject supported the hypothesis that steroid abuse was likely a perpetuating factor in the development of reverse anorexia; chronic use of steroids has been associated with significant psychiatric complications such as mood and behavioral disturbances, perceptual abnormalities, and withdrawal signs, with individuals reporting a history of FED, most frequently bulimia nervosa, but also anorexia nervosa or both [23].

Therefore, the use of AAS may be considered a risk factor for bigorexia nervosa in individuals searching for an ideal body image. However, it can also be regarded as a consequence of reverse anorexic symptoms. The practical implication of these studies is that screening for excessive AAS use should be granted to patients with confirmed bigorexia nervosa as well as populations vulnerable to this disorder.

The type of sport practiced was explored in relation to the potential vulnerability to bigorexia nervosa. Weightlifters with this disorder had significantly higher scores for body dissatisfaction, abnormal attitudes towards eating, risky use of AAS, and a higher lifetime prevalence of mood, anxiety, and FEDs when compared to healthy controls practicing the same sport [20]. Although this report did not confirm weightlifting as a definite risk factor for bigorexia nervosa, it indicated a possible correlation with body dissatisfaction. In the same study, weightlifters with muscle dysmorphia (N_1_ = 24 men) frequently presented feelings of shame and embarrassment and impaired functioning in social or professional areas compared to those without (N_2_ = 30 men) [20]. The causal relationship between other psychopathologies and bigorexia nervosa has not been explored. However, it can be concluded that weightlifters with bigorexia nervosa could benefit from screening for other psychiatric symptoms.

Another potential vulnerability factor explored in bigorexia nervosa was the type of faculty followed. Clinical features suggesting the presence of bigorexia nervosa were up to five times more frequent in students following Exercise and Sport Sciences classes than in those registered in Biology or Dietetics school [26]. This type of risk factor is still difficult to evaluate, and replication studies are required.

### 3.2. Positive and Differential Diagnosis

Based on the data retrieved from the literature, the core elements of bigorexia nervosa are (1) cognitive phenomena, i.e., obsessions about the smallness and weakness of one’s own body, and (2) behavioral symptoms, such as excessive physical exercise and changes in diet +/− AAS abuse [13,14]. It is considered either an FED, a part of the obsessive–compulsive disorder (OCD) spectrum, or a variant of BDD, depending on the importance placed by different authors on the core symptoms of the disorder (obsessive thoughts or prevalent ideas related to eating, body shape, and weight are also seen in anorexia nervosa; the nature of these ideas is close to that of obsessions; the content of these obsessions is, nevertheless, body image) [13,14]. Several authors have suggested that muscle dysmorphia may be considered an addiction to body image because of the efforts of individuals with this disorder to maintain a pre-defined body ideal by any means possible [27].

Individuals’ insights into their cognitive and behavioral symptoms vary [19]. They tend to adhere to a strict diet based on high-protein and low-fat foods with a clearly pre-defined calorie content per day, and they become very anxious if they unintentionally deviate from these rules [19]. In addition, they may refuse to change their exercise program or diet, even if they are aware of the possible negative medical consequences [19]. A meta-analysis targeting the available diagnostic criteria for muscle dysmorphia found evidence supporting the existence of this disorder as a clinical entity [28]. However, this review did not find sufficient data to clarify whether it may be better considered a specifier of BDD or a unique, distinct diagnosis [28].

In order to explore the semiologic differences between BN and BDD, an analysis of the available diagnostic criteria was conducted, using as references the two most influential classifications of psychiatric disorders developed by the American Psychiatric Association [21] and the World Health Organisation [22]. According to the fifth edition of the Diagnostic and Statistical Manual of Mental Disorders (DSM-5 TR), BDD with muscle dysmorphia is a condition included in the OCD spectrum [21]. (Figure 2). The core diagnostic feature of this disorder is the preoccupation with the idea that the body is too small or lacks muscular mass; this defect is not observable or appears slight to others; at some point during the disorder, the patient has performed repetitive activities or mental acts in response to appearance concerns; there is a degree of clinically significant stress that impairs social, occupational, or other important areas of functioning; and the intense interest in one’s own appearance is not better explained by concerns with body fat or weight in the presence of a diagnosis of the eating disorder [21]. The insight of the patient regarding his condition can vary from “good or fair” to “absent”, and delusional beliefs may be associated with it [21]. The core interest of the patient may be related not only to muscle mass but also to hair or skin [21]. The special type of “BDD by proxy” can include muscle dysmorphia, in which individuals are preoccupied with defects they perceive in the appearance of others [21].

The 11th edition of the International Classification of Diseases (ICD-11) defines muscle dysmorphia as a variant of BDD that usually affects men and increases their risk for somatic complications such as muscle tears, strains, and adverse events of steroid use [22]. The core diagnostic criteria for BDD are similar to those mentioned in the DSM-5, and it is part of the chapter dedicated to “obsessive compulsive and related disorders” [22]. Ideas of self-reference can be reported by these patients with a focus on their perceived body deficits [22]. Insight can be fair to good or poor to absent [22].

Regarding the differential diagnosis of BDD within the frameworks of the DSM-5 TR and ICD-11, it should be noted that, unlike FEDs, patients with this pathology do not have preoccupations exclusively related to body image, and they may be concerned with ”one or more perceived defects or flaws in appearance that are either unnoticeable or only slightly noticeable to others”. These patients may engage in unusual eating behaviors, such as excessive protein or nutritional supplements, or participate in excessive exercise because they want to become more muscular, not because they want to maintain a low body weight [22].

Other important conditions that must be differentiated from BN are social anxiety disorder, psychotic disorders, and major depressive disorders. *Social anxiety disorder* is centered around the potential criticism that others may manifest as a result of the patient’s presumed behavioral manifestations of anxiety, while in BDD, the focus is on the evaluation of one’s own flawed physical appearance by others. *Psychotic disorders* may include delusions about one’s own body weight or shape, and the concerns observed in patients with BDDs are replaced by ideas that are impenetrable to counterarguments; however, BDD may be associated with a lack of insight into its most severe forms. Other OCDs include recurrent thoughts not limited to one’s own appearance and diverse compulsions intended to decrease these obsessions. *Major depressive disorder* can be associated with preoccupations or worries related to imaginary physical flaws or defects; however, these thoughts are part of a more complex clinical picture, such as anhedonia, depressed mood, insomnia/hypersomnia, loss of/increase in appetite, etc.

Also, BN is distinct from *bodily distress disorder*, which includes multiple somatic symptoms that may have a variable evolution, with excessive attention directed toward them, although appropriate clinical examinations and investigations have failed to identify specific diseases [22]. *Persistent bodily symptoms* have a significant impact on the patient’s personal, family, social, educational, occupational, or other areas of functioning [22]. *Concern about body appearance* is a residual code in ICD-11, included in the category of “reasons for contact with the health services” [22], but the intensity and functional impairment are clearly not significant to justify a diagnosis of BDD or other well-defined nosological entities. 

*Behavioral addictions*, including physical exercise addiction, are currently explored as important reasons that lower the quality of life and worsen the prognosis of comorbid psychiatric disorders [29]. Unfortunately, there is still no consensus on what constitutes “excessive” exercise as the core criterion of addiction, because of the multiple variables involved in this construct [29,30,31]. Unhealthy exercise can be considered a core feature of muscle dysmorphia [29], but in behavioral addiction, the focus is on the reward that the exercise provides itself, not on its beneficial effects on muscle mass. 

### 3.3. Epidemiology

Data regarding the epidemiology of bigorexia nervosa are quite sparse; however, due to the increased popularity of bodybuilding and increased access to gymnasium centers, this condition may become a health problem with social significance [13]. In a previous study, 8.3% of the 108 bodybuilders were found to have “reverse anorexia” [16]. In another study, which included participants from the Dietetics School, Exercise and Sport Science School, and Biology School (N = 440 individuals, in total), the mean prevalence of muscle dysmorphia was 5.9%, based on the completion of Muscle Dysmorphic Disorder Inventory scores [26].

Based on a literature review (n = 34 articles), the lifetime prevalence rates of muscle dysmorphia in male weightlifters were between 13.6% and 44%, with variations in demographic variables and diagnostic criteria [32]. Another systematic review identified 5% (in women) and 15% (in men) prevalence of muscle dysmorphia in military personnel and associated problems such as excessive bodybuilding, use of anabolic steroids or stimulants, weight- and shape-related preoccupations, and dysfunctions associated with the desire to control body weight [33].

### 3.4. Structured Evaluation

Three instruments dedicated to the measurement of bigorexia nervosa manifestations were identified during the literature search. The *Drive for Muscularity Scale (DMS)* is self-administered and comprises 15 items; it was administered to 197 adolescents for initial validation [34] and detected a higher drive in boys. Adolescents, regardless of gender, were associated with engaging in more intensive training and following a diet designed to gain weight [34]. According to the authors of this instrument, it was created based on a list of motivations obtained from people involved in weight training and assesses “people’s attitudes about their muscularity and motivation to become more muscular” [34].

The *Male Body Check Questionnaire (MBCQ)* is also self-administered with 19 items and 4 subfactors for measuring different body-checking behaviors [35]. These factors are “global checking”, “chest and shoulder checking”, “other-based checking”, and ”body testing” [35]. Scores on this instrument correlated positively with perfectionism, FED psychopathology, and muscle dysmorphia symptoms [35]. The authors recommend that this questionnaire be used to “investigate body image-based pathology in males” [35]. The *Muscle Dysmorphic Disorder Inventory (MDDI)* comprises 21 items and assesses manifestations of bigorexia nervosa through its three dimensions: “drive for size”, questions about the desire to be smaller, less muscular, and weaker than desired, and the wish to increase in strength or body size; “appearance intolerance”, questions about negative beliefs related to one’s own body, anxiety about body size, and body exposure avoidance; and “functional impairment”, questions referring to excessive exercise, negative emotions when routine exercise is missed, and avoidance of social exposure due to negative feelings and preoccupations with own body [36].

Considering the characteristics of these instruments, their recommended use is screening for bigorexia nervosa in vulnerable populations. Note that MBCQ and MDDI were validated in male individuals; therefore, the extrapolation of their results to females is not supported by evidence. Screening vulnerable populations for detection in the early phases of the disorder could be the best way to prevent the onset of somatic or psychiatric complications (e.g., muscle tears, AAS abuse, and depression or anxiety symptoms).

### 3.5. Available Treatments for Bigorexia Nervosa

Psychotherapy, especially cognitive behavioral therapy (CBT), focuses on the cognitive restructuring of obsessive thoughts related to body image and perfectionism, while dialectical behavioral therapy (DBT) focuses on training emotion regulation skills. The latter is considered beneficial, but the evidence supporting this indication is sparse [37]. In a randomized study (N = 79 male collegiate athletes), a 3-session group intervention increased satisfaction with specific body areas and decreased several risk factors for FEDs, that is, valuing muscularity, self-administration of supplements, and idealizing body shape/weight vs. the control group [38]. According to a meta-analysis (n = 79 trials), therapist-led CBT was more efficacious than wait lists and any active comparator (other psychotherapies) in patients with bulimia nervosa and binge eating disorders [39]. Data supporting the use of CBT in anorexia nervosa patients are limited, although they can be useful in improving some key outcomes (e.g., BMI, eating disorder symptoms, and broader psychopathology), although these results were not consistently superior to other treatments (e.g., dietary counseling, interpersonal therapy, etc.) [40].

Family therapy may be helpful in adolescents with muscle dysmorphia; however, this recommendation was based on a single case study [41]. In this particular case, a 15-year-old boy presenting behavioral and cognitive features of muscle dysmorphia enrolled in a 10-session family therapy spanning seven months [41]. The core psychotherapy concepts for this patient were derived from a family-based approach to AN treatment [41]. Although the results were favorable, that is, remission of muscle dysmorphia symptoms was attained, there is an obvious need to replicate these data to support the recommendation of family therapy for this specific pathology.

No data are available on the efficacy of pharmacological treatments for this condition. However, from a case-management perspective, it is essential to properly screen for comorbid pathologies, especially AAS addiction, because these conditions need to be treated concomitantly.

## 4. Discussion

Bigorexia nervosa represents a controversial nosological construct due to its complex clinical characteristics, which bear important practical consequences, such as the need to clarify its definition in order to formulate adequate prevention strategies and validate adequate therapeutic interventions. Based on the explored data, bodybuilders, weightlifters, and other populations involved in athletic activities are most vulnerable to the onset of this disorder [13,25,27]. Whenever cases of bigorexia nervosa are identified, patients should also be screened for physical and psychiatric comorbidities and complications, such as AAS use disorder, physical exercise addiction, other behavioral/substance use disorders, and mood disorders. This recommendation is supported by a large body of data reviewed in the current paper [14,20,23,24,25,27,33,38]. The diagnosis criteria used for characterizing clinical manifestations of bigorexia nervosa are not yet unanimously accepted, highlighting the controversies around this nosological entity. While both the DSM-5TR and ICD-11 consider it as a BDD variant, these classifications only describe in a sentence or two its core features [21,22]. Additionally, various authors constructed their own sets of criteria for diagnosing bigorexia nervosa or muscle dysmorphia, adding more complexity to the definition of this concept [19,21,28]. It is important to bear in mind that bigorexia nervosa has manifestations of both BDDs and Eds; therefore, clinicians and researchers wishing to investigate this pathology should consider a larger nosological perspective than the one suggested by current systems of mental disorder classification. Therefore, positive and differential diagnoses should include a variety of psychiatric disorders (including behavioral addictions, with which bigorexia nervosa may be related) and also the preoccupations with physical appearance that may be detected in otherwise healthy individuals [27,42]. Also, the differential diagnosis is important to delineate the core features of bigorexia nervosa from those of schizophrenia spectrum disorders with somatic delusions and ideas of reference, depressive disorders with self-image distortions, and SUDs accompanying mood or anxiety disorders [43,44,45,46].

Although specific data on the pathophysiology of bigorexia nervosa are extremely scarce, it may be concluded, based on the retrieved data, that this disorder appears as the result of combining a vulnerable background (e.g., AAS use disorder, type of faculty graduated from, self-image distortions, psychological rigidity, and type of sport practiced) [14,20,24,25,26] with insufficiently explored psychosocial and cultural triggers (e.g., social pressure about an idealized body shape, dysfunctional communication between family members, depression, and shame) [1,2,3,4,5,9,10,19]. The cultural model of a male body shape defined as “bigger, bulkier, and more muscular” [34] may induce, especially in adolescents and young individuals, a drive for excessive exercise, AAS use, and an exaggerated focus on their own body shape. Therefore, an implicit aim of this review was to increase awareness about the existence of bigorexia nervosa in mental health specialists who are involved in evaluating such vulnerable populations— adolescents, bodybuilders, military personnel, patients with SUDs or behavioral addictions, etc.

Three validated psychological tools that may evaluate in a structured manner the BN clinical manifestations have been found in the literature [34,35,36]. MDDI, MBCQ, and DMS may be administered for screening in vulnerable groups, but the first two instruments were not validated on the female population. Therefore, caution is needed when interpreting the results of these tests in a mixed male and female population.

Therapeutic interventions for patients with bigorexia nervosa are still in the early phase of development; however, based on the similarity with the OCD spectrum, CBT and SSRI administration have been suggested. Therefore, whether defined as a variant of BDD or part of the OCD spectrum, a trial with selective serotonin reuptake inhibitors (SSRIs) seems to be granted. SSRIs have been associated with good results in both OCDs and some cases of FEDs, although negative studies on patients with anorexia nervosa exist [6,47,48,49,50,51]. Both CBT and SSRIs are recommended as first-line approaches for BDD [52].

This research has several practical implications, such as how (a) available data support the need for revising the current classification of bigorexia nervosa as a BDD variant based on data about clinical characteristics, alternate diagnosis criteria, and risk factors; (b) several domains of BN research still require extensive investigation, such as the existence of BN in females, the pathophysiological substrate of this disorder, or the validated therapeutic options; (c) the identified risk factors and presumed etiological factors of bigorexia nervosa can be integrated into a new screening tool, such as a structured interview that may be useful for screening and monitoring purposes. The principal limitation of this review was the lack of a systematic methodology. The narrative strategy was preferred because the aim of this study (as mentioned in its two operationalized objectives) was to increase awareness among psychiatrists and psychologists about the risks of bigorexia nervosa/muscle dysmorphia. However, due to the lack of a systematic approach, it is possible that some reports may not have been included in the current review [53,54,55]. Also, no assessment of the quality of the research was conducted due to the fact that only one author was involved; therefore, all sources were considered of equal importance for the objectives of the research. The strength of this review consists of its focus on the five dimensions that are frequently used for the characterization of any illness, i.e., the epidemiology, criteria for positive and differential diagnosis, risk factors, methods of evaluation, and therapeutic approaches. Due to this structure, the areas needing further exploration and the domains with consistent data to support the existence of bigorexia nervosa can be more easily detected.

Future research directions are represented by the need to identify genetic vulnerability factors, the characteristics of bigorexia nervosa in the female population, the impact of specific environmental factors that modulate this potential vulnerability, the types of prevention programs that may be implemented in vulnerable populations, treatments that are adequate for patients with bigorexia nervosa, and the duration of their administration.

## 5. Conclusions

At the crossroads between FEDs and BDDs, with a history of more than three decades, the concept of BN still requires extensive research in order to define its clinical characteristics and therapeutic management. However, the current review found 26 reports that support, in different ways and with different strengths, the existence of this disorder. It might be expected that further research would change the perspective of bigorexia nervosa as a variant of BDD and allow its existence as a distinct disorder that integrates elements from both anorexia nervosa and BDD. To attain this objective, larger studies in more diverse populations are needed, as well as the development of new, good-quality psychological instruments able to detect the specific features of this disorder in the general population. At the same time, it is important to avoid over-pathologizing healthy individuals with a transient concern about their own physical status, who are exercising within normal limits, and who are taking a cautious approach to their eating habits.

## Figures and Tables

**Figure 1 brainsci-13-01234-f001:**
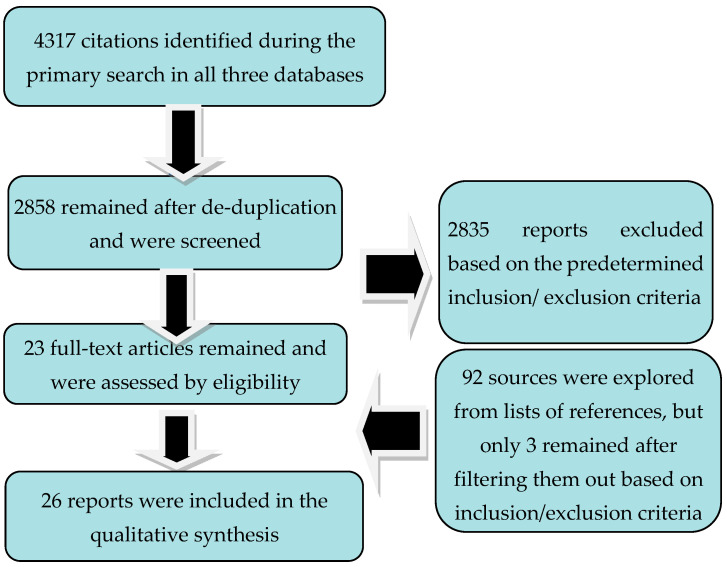
Flowchart of reviewing process.

**Figure 2 brainsci-13-01234-f002:**
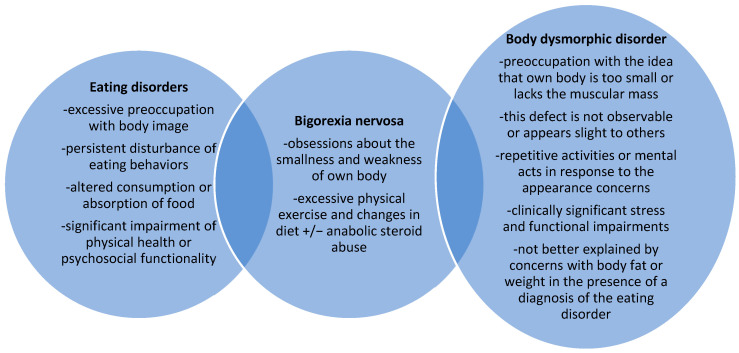
Bigorexia nervosa at the crossroads between eating disorders and body dysmorphic disorders.

**Table 1 brainsci-13-01234-t001:** Inclusion and exclusion criteria for the review.

Operational Criteria	Inclusion Criteria	Exclusion Criteria
**Population**	All age groups were allowed. The diagnoses explored were “bigorexia (nervosa)”, “muscle dysmorphia”, or “reverse anorexia”. Diagnoses made according to the DSM or ICD nosographic systems (no limitations regarding the edition) were permitted, but also original criteria constructed by authors of the respective reports.	Unspecified diagnoses or reports that included various EDs or BDDs without clarifying what criteria were used during the research.
**Intervention**	Any type of study, such as clinical or preclinical research, epidemiological or clinical, prospective or retrospective, etc. Any type of review, such as systematic, narrative, scoping, meta-analysis, umbrella review, etc.	Studies with undetermined methodology and reviews with unspecified design.
**Environment**	Inpatient, outpatient, daycare, and general population.	Unspecified environment.
**Primary and secondary variables**	Prevalence, incidence, risk factors, clinical diagnosis, pathophysiological data, psychological evaluation, and treatment.	Imprecisely defined or poorly characterized variables and reports without pre-defined outcomes.
**Study design**	Primary and secondary reports, clinical and preclinical research.	Unspecified or insufficiently defined designs.
**Language**	Any language of publication was admitted if the in extenso version of the paper could be retrieved.	

ED = eating disorder, BDD = body dysmorphic disorder, DSM = Diagnostic and Statistical Manual of Mental Disorders, and ICD = International Classification of Diseases.

**Table 2 brainsci-13-01234-t002:** Summary of the reviewed reports on bigorexia nervosa.

Reference	Type of Paper	Main Outcomes	Results and Observations
[13]	CR, a male practicing BB	Pathophysiological considerations about BN, bodybuilding as a predisposing factor to BN, analysis of BN as a type of ED	Psychological vulnerability for BN—a young man reports experiencing concerns about his appearance or muscularity. Predisposing factors—physical exercise motivated primarily by the need to improve one’s own physical aspect.
[14]	A case–control study (AAS-abusing males practicing BB vs. non-AAS-abusing males practicing BB), N = 108 participants	SCID-III-based evaluation for detection of AN, Bn, BN + medical history, family history of psychiatric disorders and violent behavior + physical examination	BB is a risk factor for BN and AN. BN may precipitate or perpetuate AAS use in vulnerable individuals.
[16]	Retrospective, case–control study (BDD + BN vs. BDD without BN), N = 63 patients	The detection of characteristics in patients with BDD + BN at the clinical level	Demographic variables, BDD severity, delusionality, and number of non-muscle-related body parts of concern were similar between the two groups. Attempted suicide, poorer quality of life, and higher rates of SUDs and AAS abuse were more frequent in the BN + BDD group.
[19]	NR + CR (four patients—two male and two female)	Diagnostic features, differential diagnosis, epidemiology, and etiology	Tentative diagnosis criteria are suggested, and case reports are presented.
[20]	A case–control study (men with BN vs. healthy weightlifters), N = 54 participants	Demographic, psychiatric, and physical measures	BN was associated with body dissatisfaction, dysfunctional eating attitudes, higher prevalence of AAS use, and higher lifetime prevalence of mood disorders, anxiety disorders, and ED vs. controls.
[21]	Nosological classification	Diagnostic and epidemiological data on muscle dysmorphia	BN is a subtype of BDD, and it is included in the OCD spectrum. Specific diagnostic criteria were formulated.
[22]	Nosological classification	Diagnostic data on muscle dysmorphia	BN is a form of BDD.
[23]	NR	Relationship between AAS use and BN	AAS use is considered secondary to the need to compensate body image pathology, BN included. AAS use is frequently detected in patients with BN; therefore, this SUD is a part of the diagnostic criteria for BN.
[24]	A case–control study (AAS users vs. non-users), N = 89 male weightlifter participants	Relationship between AAS use and body image distortions and attitudes toward male roles	Body image pathology and rigid stereotypic views on masculinity are prominent in men with long-term AAS use. However, the causality is still uncertain.
[25]	A case–control study included current, ex-, and non-AAS BB, N = 137 male participants who practiced regular aerobic exercises	Relationship between AAS use and reverse anorexia	AAS users had an ideal image of an exaggerated mesomorphic body shape. Current and ex-AAS users had higher scores on eating behavior pathology vs. non-AAS users. AAS use, but not BB as an independent variable, was associated with higher severity of BN.
[26]	A case–control study (students in Dietetics/Exercise and Sport Sciences vs. Biology), N = 440 participants	Prevalence of orthorexia and BN in students following courses on nutrition and body care vs. those following Biology classes	The selection of the university courses may be, at least partially, under the influence of eating behavior pathology.
[27]	NR	Exploration of an alternative model for BN—”the addiction to body ideal” paradigm	Maintaining a body image through bodybuilding, exercise, eating behaviors, AAS use, etc., is addictive; therefore, BN may be considered a behavioral addiction.
[28]	SR and MTA (n = 40 papers)	Clinical diagnosis of BN	Using cluster analysis, two distinct patterns of BN were identified—the cognitive type and the behavioral variant. The data analyzed did not prove to be sufficient for determining if BN is a specifier for BDD or a unique psychiatric disorder.
[29]	SR protocol	Exploration of compulsive exercise in EDs and BN	Not yet disclosed.
[30]	NR	Exploration of compulsive exercising in EDs, especially restrictive type AN	Compulsive exercising is a negative prognosis factor in EDs, and it is associated with higher ED psychopathology, dietary restraint, general psychopathology, and personality features (perfectionism, persistence, and lower novelty seeking). Enhanced CBT may be useful for hospitalized patients with this pathology.
[31]	Delphi study (N = 25 participants, experts in AN treatment)	Exploration of unhealthy exercise in adolescents with AN	Exercise restriction practices and initiating healthy exercise behaviors in patients who are medically stable.
[32]	SR (n = 34 papers)	Clinical criteria for diagnosing BN	There is insufficient data to support the validity, clinical utility, nosological classification, and inclusion of BN as a new disorder in the nosological systems of mental disorders.
[33]	SR (n = 20 papers)	Analysis of BDD, BN, weight and shape dissatisfaction, and the use of AAS in military personnel	BDD, BN, and AAS use are highly prevalent in military personnel. A preliminary tendency toward overvaluation of physical appearance and fitness may influence the decision to choose a military career.
[34]	Validation study for a psychological instrument	Creating and validating a psychometric tool to determine the severity of BN manifestations	A 15-item survey was administered to 197 adolescents, and the drive for muscularity displayed good reliability. This drive was related to lower self-esteem and a higher level of depression in boys, but not in girls.
[35]	Validation study for a psychological instrument	Convergent and divergent validity, factor structure, and reliability of a psychometric tool for BN	A 19-item, four-factor instrument was supported by analysis + a higher-order global checking factor for men only. Good concurrent and divergent validity and good short-term test–retest reliability.
[36]	Validation study for a psychological instrument	Psychometric properties of a new instrument for quantifying BN severity	A 13-item questionnaire with two subscales had good reliability, convergent and divergent validity, and test–retest validity.
[37]	NR	Overview of clinical features and treatment options for BN	Techniques applied with success for treating BDDs and EDs can be intuitively used for patients with BN. Cognitive restructuring of perfectionistic and egosyntonic beliefs and also DBT techniques may prove useful.
[38]	A case–control study (psychotherapy vs. assessment-only), N = 79 male participants	The effects of a psychotherapeutic, structured intervention (3-session group Male Athlete Body Project) in collegiate athletes with EDs	Satisfaction with specific body parts increased, and the drive for muscularity and body-ideal internalization decreased at post-treatment evaluation vs. controls. The results were preserved at the 1-month follow-up visit. Reductions in AAS use were also observed.
[39]	MTA (n = 72 trials)	Efficacy of CBT for EDs	CBT is efficacious for patients with EDs, and it outperformed all active psychological comparisons and ITP, specifically. Many trials were of poor quality.
[40]	SR (n = 16 articles)	Efficacy of CBT for AN	CBT improved treatment adherence and reduced the dropout risk in patients with AN. CBT was not significantly superior to dietary counseling, supportive management, ITP, or behavioral family therapy.
[41]	CR, a 15-year-old boy	The effects of family-based treatment for BN	This intervention may be efficient. MDDI score decreased significantly at the post-treatment visit vs. baseline.

AAS = anabolic steroids; AN = anorexia nervosa; BB = bodybuilders; BDD = body dysmorphic disorder; CBT = cognitive behavioral therapy; CR = case report; Bn = bulimia nervosa; BN = bigorexia nervosa; DBT = dialectical behavioral therapy; ED = eating disorder; ITP = interpersonal therapy; MDDI = Muscle Dysmorphic Disorder Inventory; MTA = meta-analysis; NR = narrative review; OCD = obsessive–compulsive disorders; SCID = Structured Clinical Interview for DSM-III; SR = systematic review; and SUD = substance use disorder; N was intended for number of patients, while n was used for number of trials.

## Data Availability

This review did not explore new data but synthesized the results of the previously published reports.

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
