# Peer review of "At the Crossroads between Eating Disorders and Body Dysmorphic Disorders—The Case of Bigorexia Nervosa"

_brainsci, 2023, doi:10.3390/brainsci13091234_

Round 1
Reviewer 1 Report
Dear Authors,
Thanks for the review. The aim was to explore the characteristic features of Bigorexia nervosa and to identify evidence-based therapeutic interventions for this condition. The topic is relevant because it is necessary to clarify the concept of Bigorexia nervosa, to develop adequate prevention strategies and to valid proper therapeutic interventions.
Some suggestions for improving the publication:
1. The methodological part is written very superficially, there is a lack of information to make it possible to repeat the mentioned research. How many articles were selected? What were the exclusion criteria?
2. Why was the database Google Scholar chosen instead of, for example, Scopus or Web of Science where articles from high-ranking journals are available?
3. The discussion part is very poor. I would recommend expanding the discussion part, considering that the authors have studied a number of publications on the mentioned topic, they have formed their own opinion, which would be desirable to expand in the discussion.
Author Response
Thank you very much for your thorough review and very useful observations. Please note my answers below:
Q1. The methodological part is written very superficially, there is a lack of information to make it possible to repeat the mentioned research. How many articles were selected? What were the exclusion criteria?
A1: Thank you for pointing this out, the methodological section was expanded, and a new table regarding the inclusion/exclusion criteria was added (Table 1). Also, a new figure (Fig.1) was included, to illustrate the selection process of the reviewed sources.
Q2. Why was the database Google Scholar chosen instead of, for example, Scopus or Web of Science where articles from high-ranking journals are available?
A2: Indeed, Google Scholar was a second choice, but I still preferred it because the subject of bigorexia nervosa is not extensively researched, and grey literature may have been important for finding more reports on this topic.
Q3. The discussion part is very poor. I would recommend expanding the discussion part, considering that the authors have studied a number of publications on the mentioned topic, they have formed their own opinion, which would be desirable to expand in the discussion.
A3: The “Discussion”, as well as the “Conclusion” sections were re-structured, according to your very helpful recommendation.
Reviewer 2 Report
This is interesting research that aims to explore the characteristic features of Bigorexia Nervosa and identify evidence-based therapeutic 10 interventions for this condition.
I will try to let the authors know some of the improvements they could add to the manuscript. However, it will not have been easy research and I congratulate them on having carried it out. Below are my comments
- Well-structured summary
-Line 55 and 72 check inverted commas and their position.
- Before commenting on section 2, consider whether to include a paragraph that ends the introduction and clarifies again the purpose of the review in order to justify that it is to be carried out.
- The process of searching for information, key words, manuscripts found and filters for discarding them, etc. should be better clarified.
- The discussion is extremely short for all the information that has been collected. It should be considered to restructure the article and better compare all the information that appears in the results in order to divide it and include it in the discussion with all its references.
- Conclusions section practically undeveloped
The information provided is relevant but the way it is structured is wrong in my opinion. In the discussion, as I have pointed out, the results should be commented on and compared. Moreover, it is a section that should be cited in order to make those comparisons and discuss them. The section describing the search carried out for the review should also be significantly improved. On the other hand, I would advise authors to include a table naming the most relevant references and authors of this search as well as including a section on practical applications.
The manuscript requires significant improvements for publication.
Author Response
I wish to thank the Reviewer very much for his/her thorough review, very kind words, and most useful observations. Please note my answers below:
Q1: Well-structured summary
A1: Thank you, I have added information about methodology and delineated the conclusive section of the “Abstract”.
Q2: Line 55 and 72 check inverted commas and their position.
A2: All the inverted commas were verified and replaced with the correct signs throughout the text.
Q3: Before commenting on section 2, consider whether to include a paragraph that ends the introduction and clarifies again the purpose of the review in order to justify that it is to be carried out.
A3: Lines 77-90 were added according to your very useful recommendation.
Q4: The process of searching for information, key words, manuscripts found and filters for discarding them, etc. should be better clarified.
The section describing the search carried out for the review should also be significantly improved.
A4: The methodological section was expanded, and Table 1 and Fig.1 were added.
Q5: The discussion is extremely short for all the information that has been collected. It should be considered to restructure the article and better compare all the information that appears in the results in order to divide it and include it in the discussion with all its references.
A5: The “Discussion”, as well as the “Conclusion” sections were re-structured.
Q6: Conclusions section practically undeveloped. The information provided is relevant but the way it is structured is wrong in my opinion. In the discussion, as I have pointed out, the results should be commented on and compared. Moreover, it is a section that should be cited in order to make those comparisons and discuss them.
A6. Indeed, these two sections necessitated further development, and they were updated, according to your very helpful suggestions.
Q7: On the other hand, I would advise authors to include a table naming the most relevant references and authors of this search as well as including a section on practical applications.
Q7: Table 2 was added to support the information presented in Chapter 3. A paragraph on practical applications was added in the “Conclusion” section.
Round 2
Reviewer 2 Report
The authors did an excellent job revising the manuscript. They have incorporated all suggestions and improved the article's quality. Including the table of authors is an invaluable addition. However, I believe it does not adhere to the journal's format requirements. If the authors revise this section of the manuscript, I believe it can be accepted for publication.
Author Response
Thank you very much for the observation regarding the format requirements for the tables. I have adjusted these elements according to the journal's model.